# Modeling magnetopause location for 4D drift-resolved radiation belt codes: Salammbô model implementation.

Rabia Kiraz[1], Nour Dahmen[1], Vincent Maget[1], Benoit Lavraud[2]

[1] ONERA/DPHY, Université de Toulouse, Toulouse, F31055, France

[2] Laboratoire d'Astrophysique de Bordeaux, Université de Bordeaux, France

*Correspondence to*: Rabia Kiraz (rabia.kiraz@onera.fr)

**Abstract.** We present a new semi-analytical magnetopause location model specifically designed for 4D drift-resolved radiation belt modeling codes. We specifically designed this magnetopause location model for the 4D version of Salammbô but can be adaptable to similar codes. The model combines parameterization by the $Kp$ index with a representation of the magnetopause

in $L^*$ geomagnetic coordinates and magnetic local time (MLT). It is based on a 20-year dataset relying on computed magnetopause stand-off distances using a solar wind database and a relevant magnetopause model, then converted into $L^*$ geomagnetic coordinates for all dayside MLT. Through statistical analysis of this dataset, the model was formulated and validated against a magnetopause crossing catalog. Its performance was benchmarked against the magnetopause location model previously developed for the 3D version of the Salammbô code. Results demonstrate improvement in predicting the

magnetopause position in $L^*$ across dayside MLT sectors, with enhanced accuracy in the dawn sector. These results highlight the model's ability to model the magnetopause location in $L^*$ across dayside MLT sectors. This advancement may be specifically useful for simulating magnetopause shadowing in ring current and radiation belt modeling codes.

## 1. Introduction

The magnetopause, the outer boundary of Earth's magnetosphere, represents the interface between the magnetospheric

magnetic field and the hot tenuous plasma of the solar wind. First theorized by Chapman and Ferraro (1933), this boundary delimits a protective cavity around the Earth. It shields various internal structures, such as the Van Allen radiation belts, from the solar wind. These belts are composed of charged particles confined by Earth's magnetic field. They are highly dynamic structures, responding strongly to enhanced geomagnetic activity. The magnetopause, as the boundary of the magnetosphere, plays a critical role in influencing the behavior of the radiation belts. Accurate characterization of the magnetopause is essential

for understanding and modeling this dynamic.

Modeling of the Earth's radiation belts has largely been based on physical codes (Beutier and Boscher, 1995; Bourdarie et al., 1996; Glauert et al., 2014b; Reeves et al., 2008; Subbotin and Shprits, 2009). Among them, ONERA's Salammbô code is a well-established physical code with versions for both electrons and protons (Beutier et Boscher 1995; Bourdarie et al. 1996). However, challenges remain, particularly for accurately modeling particle loss processes through wave-particle interactions

and dropouts. This paper specifically examines how magnetopause modeling may be implemented to help best reproduce the rapid and intense particle losses during specific dropouts, namely magnetopause shadowing effect, which impact a broad energy range of protons and electrons across various drift shells.

A characterization of dropouts has been first conducted by Bailey (1968), who described it as an increased electron precipitation on the dayside during a geomagnetic storm. Although the process is not fully understood, literature agrees on dropouts being related to two major mechanisms. The first involves particle precipitation into the atmosphere due to wave-particle interactions, either through Chorus waves (Morley et al., 2010) or EMIC waves (Xiang et al., 2017). The second mechanism is the magnetopause shadowing effect. This process happens when the magnetopause moves inward due to compression during geomagnetic storms. At the same time, magnetospheric Ultra Low Frequency (ULF) waves cause outward radial diffusion, pushing particles away from Earth toward the dayside magnetosphere (Degeling et al., 2013).

Several models have been developed for modeling analytically the location of the magnetopause (Fairfield 1971; Formisano et al. 1979; Holzer et Slavin 1978; Petrinec, Song, et Russell 1991; Petrinec et Russell 1996; Roelof et Sibeck 1993; Shue et al. 1997; 1998; Sibeck 1991). The majority of these models use an elliptical or parabolic function to represent the magnetopause. The magnetopause model by Shue et al. (1997, 1998) referred to in the following as the Shue model, is widely adopted for its accuracy (Shue et al., 2000) and ease of implementation. It is an analytical model that is built on a database of more than 500 magnetopause crossings observed by the ISEE 1 and 2 (Durney and Ogilvie, 1979), AMPTE/IRM (Bryant et al., 1985) and IMP 8 satellite missions. This model has also been used as a foundation for further improved models. Notably, Lin et al. (2010) developed an extended version of the Shue model that incorporates an asymmetric high-latitude magnetopause. It was also used to study the impact of magnetopause shadowing on electron radiation belt dynamics. Matsumura et al. (2011) investigated the correlation between magnetopause shadowing and outer electron radiation belt losses during geomagnetic storms using this model. The observed strong correlation proved that magnetopause shadowing is crucial for modeling the dynamics of radiation belts. Consequently, several magnetopause shadowing models have been developed for radiation belt modeling codes (Glauert et al. 2014a; Herrera et al. 2016; Wang et al. 2020). These are detailed in the next section.

The particles in the radiation belts are confined by the Earth's magnetic field that shapes their behavior. To accurately represent the motion of charged and trapped particles around the Earth, all existing codes rely on geomagnetic coordinates rather than geographic coordinates. One of the key geomagnetic coordinates is $L^*$, also referred to as the Roederer parameter (Roederer, 1967; Roederer and Zhang, 2014). The accurate modeling of the magnetopause shadowing phenomenon for radiation belt modeling codes relies on the magnetopause location given in the relevant adiabatic coordinates system. Herrera et al. (2016) characterized the magnetopause shadowing effect during geomagnetic storms for the Salammbô code as a function of $Kp$ and expressed it in terms of $L^*$. However, this research focused exclusively on the drift-averaged version of the Salammbô code (also called Salammbô 3D) (Beutier et Boscher 1995; Varotsou et al. 2005; Bourdarie et Maget 2012; Dahmen et al. 2024).

Yet, Salammbô exists in a drift resolved version called Salammbô 4D (Bourdarie et al., 1996). The latter considers the magnetic local time (MLT) dimension in order to model the transport of low energy electrons induced by magnetospheric convective electric fields. Integrating the physical processes represented in Salammbô 3D into the 4D code (including the addition of an MLT dependency) is a complex task. This challenge becomes even more pronounced when addressing magnetopause shadowing effect, which requires precise knowledge of the magnetopause location. In this study, we propose a simplified magnetopause location model designed to describe the magnetopause position in terms of MLT and $L^*$ for 4D radiation belt modeling codes. This model is detailed in section 2. Then we propose a validation of the new magnetopause location model against magnetopause crossing data in section 3 and we summarize our study in section 4.

## 2. Methodology and Model Development: From long-term sampled dataset Construction to Performance Assessment

### 2.1 Theoretical framework of the study

Modeling the Earth's radiation belt relies on the resolution of the Fokker-Planck diffusion equation (Schulz and Lanzerotti, 1974). All physical radiation belt modeling code solves this equation using adiabatic invariants such as the $L^*$ parameter, that simplifies the equation. Regardless of the level of refinement (1D, 2D, 3D, or 4D), the definition of the magnetopause in $L^*$ is essential for radiation belt modeling applications. However, higher levels of refinement lead to increased computational costs. The Salammbô code addresses this by using an $L^*$ grid with different grid sizes to balance accuracy and computational cost. The full $L^*$-grid consists of 133 points ranging from 1 to 8, while the smaller grid contains one-fourth of the points. In Salammbô 4D, which includes a supplementary coordinate, MLT, the reduced grid is the preferred option to ensure efficiency. Figure 1 illustrates the spacing between two consecutive grid points ($\Delta L^*$) as a function of $L^*$ for both the full $L^*$ grid (blue) and the reduced grid (red). The choice of the grid affects the accuracy of the model, with $\Delta L^*$ variations reflecting the intrinsic uncertainties of grid refinement in Salammbô codes. Consequently, when incorporating a magnetopause model into the Salammbô framework, it is essential that the magnetopause location model does not introduce uncertainties exceeding those inherent to the code.

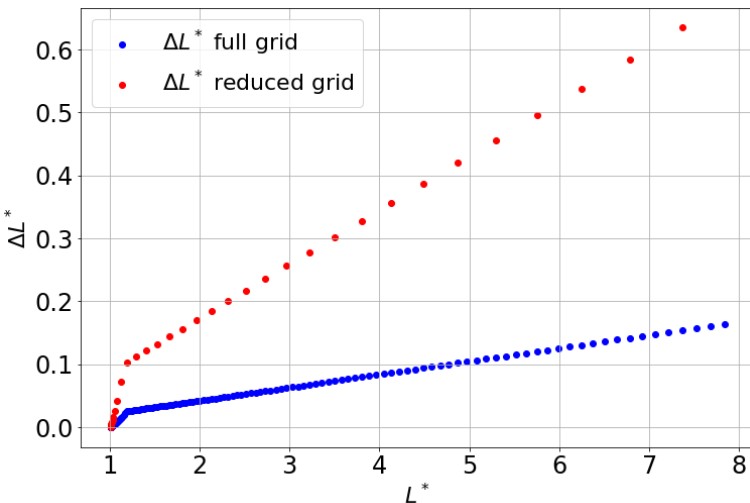

**Figure 1 : Spacing between two consecutive grid points ($\Delta L^*$) for the full $L^*$ grid (blue) and the reduced $L^*$ grid resolution (red) as function of $L^*$.**

Expressing the magnetopause location with the geomagnetic parameters $L^*$ and the MLT is crucial to ensure its adaption to radiation belt modeling codes and several examples in the literature align with this approach. For example, Glauert et al. (2014a) incorporated the Shue magnetopause model, which provides the magnetopause distance in $R_E$, into the BAS code (Glauert et al., 2014b) to compute the last closed drift shell $L^*_{LCDS}$ in $L^*$. Similarly, the VERB 3D code (Subbotin and Shprits, 2009) also incorporates a $L^*_{LCDS}$ calculation to consider the magnetopause shadowing effect in the modeling (Wang et al., 2020). Xiang et al. (2017) and Olifer et al. (2018) demonstrated that using a $L^*_{LCDS}$ calculation was more effective in capturing the magnetopause shadowing phenomena than using the distance given in $R_E$ provided by the Shue model. Complementary to these approaches, Herrera et al. (2016) proposed a quantitative estimation of the magnetopause location in $L^*$ for the Salammbô 3D code. Their method computes the magnetopause stand-off distance in Earth Radii (Re) using a magnetopause model, from which the corresponding $L^*$ value of the magnetopause ($L^*_{mp}$) is derived.

In 4D radiation belt modeling, the approach is more complex. The drift motion of particles is no longer integrated out, the modeled energy range extends to lower limits, and a transport term is added to the Fokker-Planck equation. Incorporating particle transport into radiation belt modeling means that the magnetic field is not the sole driver of particle dynamics. The magnetospheric electric fields also play a significant role in influencing particle behavior, as they can modify significantly the particle drift shells. Therefore, directly applying the 3D model of Herrera et al. (2016) to the Salammbô 4D code is not a reasonable strategy. Indeed, a magnetopause location model that integrates both $L^*$ and MLT is needed, ensuring its compatibility with 4D radiation belt simulations, while ensuring fast computation and accuracy.

Defining the magnetopause location in terms of MLT and $L^*$ presents inherent challenges. In fact, one $L^*$ value describes an entire particle magnetic drift shell, making it an integrated parameter over all MLT values. In 4D radiation belt modeling, low-energy particles orbit the Earth over relatively long time periods compared to the inward movement of the magnetopause

during periods of high geomagnetic activity (Olifer et al., 2018). Therefore, the evolution of these particle's dynamics must account for MLT, and their behavior will be influenced by the magnetospheric electric fields. As the magnetopause moves inward and intersects a drift shell, the fate of trapped particles depends on their location relative to the magnetopause boundary. Particles situated along the portion of the drift shell outside the magnetopause are lost from the system, while those on the inside remain trapped within the radiation belts. However, these particles no longer have a well-defined magnetic drift shell. Instead, their motion is likely to approximate a "virtual drift shell", guided by the combined influence of Earth's magnetic field and the magnetospheric electric fields (Burger et al., 1985; Stern, 1977).

## 2.2 Constructing the long-term sampled dataset: From Earth Radii to $L^*$

The Shue model is axisymmetric around the Sun-Earth line and provides the stand-off distance of the magnetopause in $R_E$ for each MLT value. It is driven by the z-component of the magnetic field ($B_z$) and the solar wind dynamic pressure ($D_p$), providing a link between the magnetopause location and the upstream conditions in the solar wind (Shue et al., 1998). To express the magnetopause location in terms of $L^*$ and MLT, the stand-off distance must be transformed into a $L^*_{mp}$. This transformation requires the use of a realistic magnetic field model. Many modern magnetic field models include a magnetopause boundary based on the Shue model (Tsyganenko 2002a, 2002b; Tsyganenko and Sitnov 2007). In contrast, the Tsyganenko (1989a, 1989b, 1989c) magnetic field model (referred to as T89C) is only driven by the $Kp$ index. While it differs from the other models by not having a built-in magnetopause boundary, its sole dependence to $Kp$ provides an adequate compromise between accuracy and fast computation of $L^*$ values, when compared to more sophisticated ones. Besides, its accuracy has been proven compared to in-situ measurements, even during disturbed time and large $L^*$ values (Loridan et al., 2019).

The conversion from stand-off distances to $L^*_{mp}$ comes anyway with a significant computational cost. Direct calculation of these distances within a radiation belt modeling code would considerably increase the computational time, making it impractical. To address this, we developed a long-term dataset of $L^*_{mp}$ values based on the Shue magnetopause model, covering a 20-year period. This dataset allows us to then quantify relationships between $L^*$, MLT, and dynamic parameters, reducing the need for in-code calculations.

The dataset construction process is summarized in Figure 2. It begins with the OMNI database (https://omniweb.gsfc.nasa.gov/) (represented in green), which provides measurements of solar wind dynamic pressure ($D_p$) and interplanetary magnetic field ($B_z$) values from 2000 to 2020. These values are then used to compute stand-off distances for each hourly MLT bin using the Shue model (depicted in yellow). The conversion of the stand-off distances into $L^*_{mp}$ is achieved using the $Kp$ index and the IRBEM library (O'Brien and Bourdarie, 2012) within the Python package SPACEPY (Morley et al., 2022). The conversion from $R_E$ to $L^*_{mp}$ is performed for each hourly resolved epoch at the magnetic equator. The resulting dataset (shown in blue) contains the corresponding values of $Kp$, $D_p$, $B_z$ and $L^*_{mp}$ at every dayside MLT (6h MLT to 18h MLT).

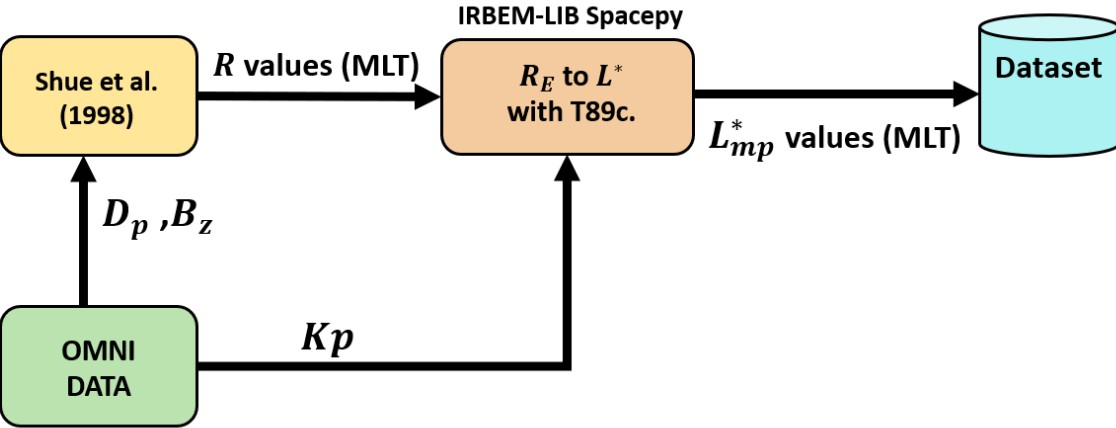

**Figure 2:** Diagram describing the long-term dataset construction process, using the OMNI data (green) to compute the magnetopause stand-off distance (yellow) and the $R_E$-to-$L^*$ transformation (orange) to build the constructed long-term dataset (blue).

### 2.3.    A semi-analytical magnetopause location model for 4D radiation belt modeling

Building a simple magnetopause model presents significant challenges, as an initial correlation analysis found no clear correlation between $D_p$, $B_z$ and $Kp$. While other studies suggest that $Kp$ is related to $D_p$ and a modified $B_z$ (Andonov et al. 2004), this adds complexity to the model development. Solar wind parameters have only been consistently available since 1995, the $Kp$ index, available since 1932, provides a reliable and long-baseline measure. Since it is the primary driver of both the T89C magnetic field model and the Salammbô code, using $Kp$ for our magnetopause model is advantageous, making it easier to integrate than solar wind parameters.

We conducted a statistical analysis on the computed 20-year dataset to investigate its dependency to the $Kp$ index. The dataset was therefore binned over four $Kp$ ranges (0-2, 2-4, 4-6, 6-9). Figure 3 illustrates the distribution of time averaged $L^*_{mp}$ values across these $Kp$ ranges for MLT sectors between 6h to 18h. For each $Kp$ range, the curves exhibit symmetry around MLT 12h. Additionally, the curves show consistent trends across the different $Kp$ ranges, suggesting that a symmetrical model driven by a single point on the curve, specifically the value at MLT 12h, can be extracted from the dataset. The values at other MLT bins can then be described as a multiplicative factor of the value at MLT 12h. Furthermore, the $Kp$ range primarily influences the shift of $L^*_{mp}$ at each MLT bin, with no significant effect on the overall shape of the curves. Figure 3 also presents the number of valid points used in the averaging for each MLT sector within each $Kp$ range. These numbers vary as nonvalid points can emerge from gaps in the OMNI database or from unsuccessful calculations of the $L^*$ parameter when the T89C models open

field lines on the given drift shell. Consequently, the MLT sector with the highest number of valid points is MLT 12, with the number of calculable points decreasing as we move further away from this sector.

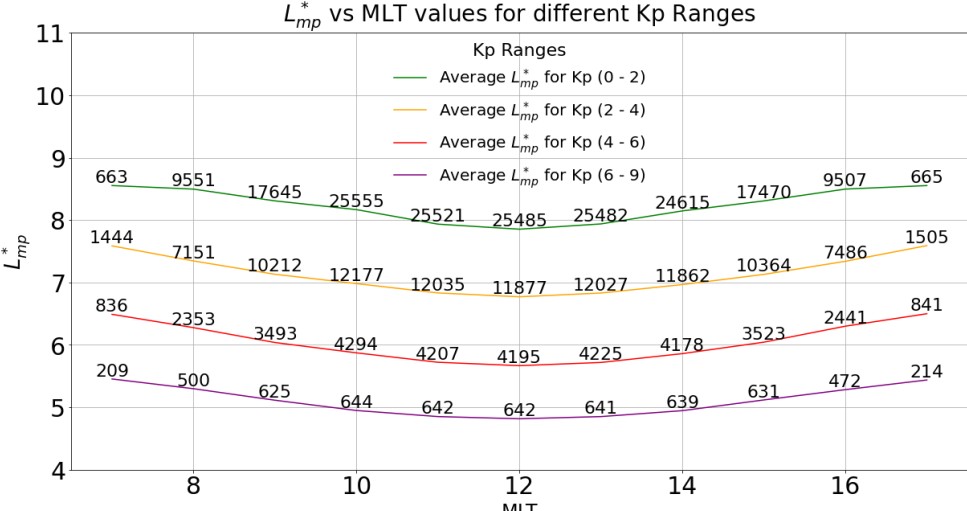

**Figure 3 :Plot of the global distribution of the dataset, with the curves representing the mean $L^*_{mp}$ values sorted per $Kp$ ranges as function of MLT values and the associated number of valid points at each MLT bin.**

The magnetopause is considered to be closest at MLT 12h, as in the Shue model. To ensure consistency, we filtered out values where the calculated magnetopause is computed as minimal outside of MLT 12h. These values arise when inconsistencies occur between the geographical position given by the Shue model and the numerical magnetic field mapping in the T89C model. These cases never exceed a few percent of the value at MLT 12h.

MLT 10h is the first MLT bin that shows a mean $L^*_{mp}$ value noticeably higher than at MLT 12h, with a difference of up to 2%.

We limited the statistical analysis to the data available at MLT 10h, ensuring that the dataset predominantly includes magnetopause locations covering MLT values between 10h and 14h. Without this limitation, the varying amount of data per MLT would have introduced bias. For example, the high concentration of values at MLT 12h, where most calculable drift shells are located, could distort the results for other MLT bins, resulting in a model that performs well at MLT 12h but is less accurate at other MLT bins. Although this approach reduces the size of our dataset, it enhances the significance of the analyzed

data. As previously established, the $L^*_{mp}$ value at MLT 12h serves as a driver for the magnetopause location model. The constructed dataset was thus normalized using this value. Consequently, the updated dataset now consists of the value of $L^*_{mp}$ at all dayside MLT bins, normalized by the $L^*_{mp}$ at MLT 12h, for epochs where a valid point exists at MLT 10h.

Figure 4 depicts the mean normalized $L^*_{mp}$ (noted $\bar{L}^*_{mp}$) and percentile distributions for each $Kp$ range as a function of MLT.

The plot depicts in gray the MLT regions below MLT 10h, where variations driven by the $Kp$ index are significant. The distribution is flat around MLT 12h with at most a deviation under 2% at MLT 11h (and MLT 13h) for all $Kp$ values and the

$\overline{L}^*_{\text{mp}}$ values at MLT 11h and MLT 13h have a maximum around 1%. This can be interpreted as the magnetopause aligns with a specific drift shell in MLT sectors close to noon. Consequently, our model considers this assumption so that for MLT [11,13]:

$$L^*_{mp} = L^*_{mp}(MLT = 12) \tag{1}$$

Additionally, the percentiles show that the $Kp$ index has a significant impact on MLT bins below MLT 10h with the $\overline{L}^*_{\text{mp}}$ equal to 4% for $Kp$ between 0 and 2, and the associated percentile reaching up to 5%. This observation suggests that the $Kp$ index affects the shape of the magnetopause and has the greatest influence on MLT values farthest from MLT 12h. Furthermore, variations in $\overline{L}^*_{\text{mp}}$ at MLT 10h are consistent with the $Kp$ dependence of the magnetopause location.

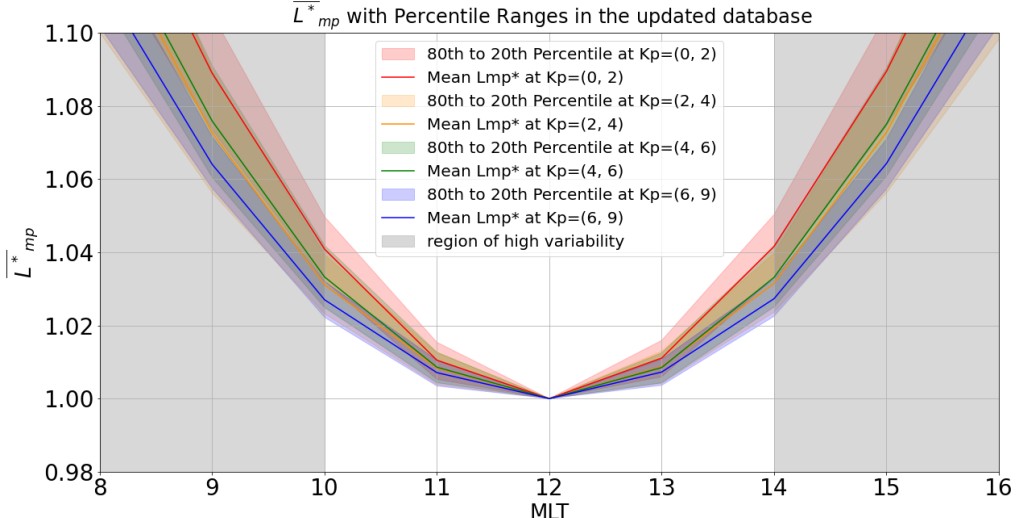

**Figure 4 : $\overline{L}^*_{\text{mp}}$ values calculated per $Kp$ ranges and their corresponding 80th to 20th percentiles, as a function of MLT with the gray area being the MLT bins of high variability.**

In order to quantify the influence of the $Kp$ index on the $\overline{L}^*_{\text{mp}}$ at MLT values between MLT 10h and 14h, we analyzed the evolution of $\overline{L}^*_{\text{mp}}$ within these MLT bins as a function of $Kp$, as shown in Figure 5. On the one hand, $\overline{L}^*_{\text{mp}}$ at MLT 11h exhibits a constant evolution as a function of $Kp$ and can thus be considered $Kp$-independent. On the other hand, the evolution of the
$\overline{L}^*_{\text{mp}}$ at MLT 10h as a function of $Kp$ can be fitted. The curve shows a "bump" at $Kp$ around 5, which has been tracked to be a numerical artifact in the $L^*_{mp}$ calculation. This artifact arises from the use of the T89C model, which introduces a $Kp$ class at 5-,5,5+, as compared to first model version (Tsyganenko, 1989a). At these radial distances and MLT bin, inconsistencies arise with other classes of $Kp$ as shown in Figure 6. Consequently, the points at these $Kp$ values have not been considered when building the regression line (Figure 5). Doing this, the evolution of the $\overline{L}^*_{\text{mp}}$ at MLT 10h is well-represented, at the first
order, by the simple linear regression in the Figure 5. The impact of the artifact on the regression has been evaluated, and it was found to have no significant effect on the regression line. This conclusion is supported by the minimal differences observed between the two regression lines, shown in red and orange in Figure 5. This outcome is reasonable, given the consistent behavior of the curve at other $Kp$ values. This regression analysis covers $Kp$ values from 0 to 8, excluding $Kp$ 9 due to the

insufficient number of data points available for that range. Nevertheless, the model is extrapolated to $Kp$ 9, ensuring the model's definition across all $Kp$ values.

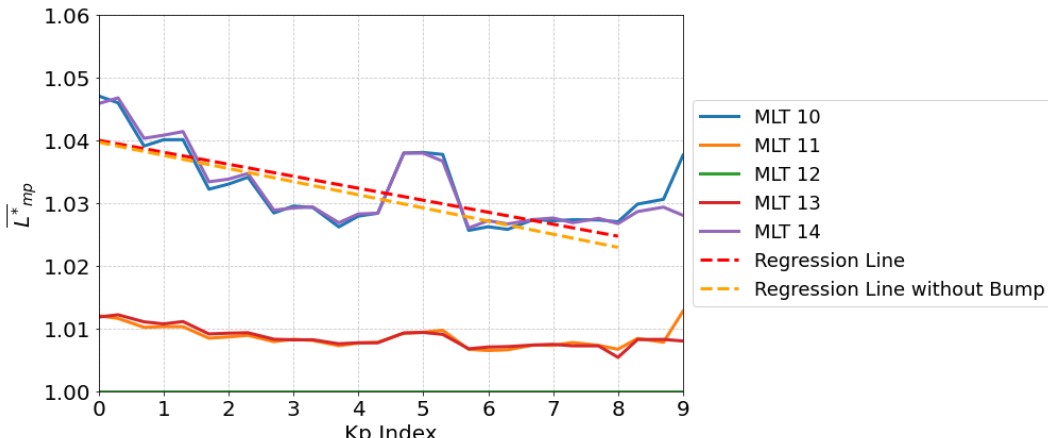

**Figure 5 : Evolution of $\overline{L}^*_{mp}$, sorted by MLT values, plotted as a function of the $Kp$ index. The linear regression lines are shown for the dataset both with and without the bump included.**

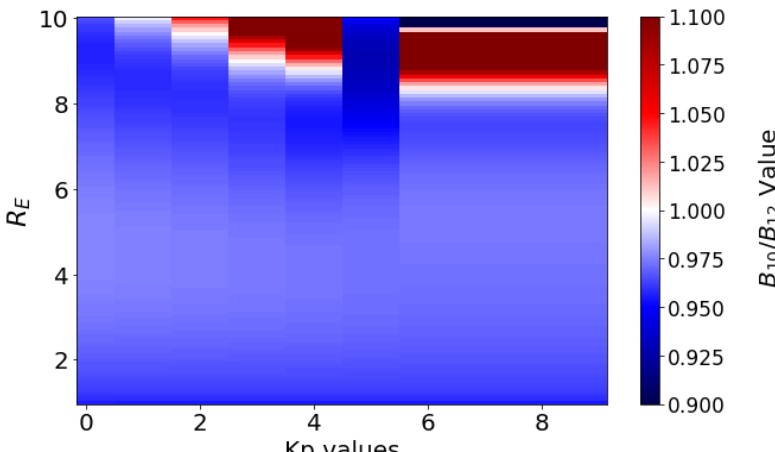

**Figure 6 : Cartography of the T89C magnetic field model. The y axis represents the radial distances and the x axis represents the $Kp$ index values. The color bar shows for the ratio of the value of B at MLT 10h to the value of B at MLT 12h.**

The function of $\overline{L}^*_{mp}$ at MLT 10h is given by Eq. (2) and its link to the $L^*_{mp}$ value is given by Eq. (3).

$$\overline{L}^*_{mp}\ (MLT = 10) = -0.00188 \times Kp + 1.0378 \tag{2}$$

$$L^*_{mp}(MLT = 10) = \overline{L}^*_{mp}\ (MLT = 10) \times L^*_{mp}(MLT = 12) \tag{3}$$

Equations (2) and (3) provide the value of $L^*_{mp}$ at MLT 10h based on the $Kp$ index. To extend the model to MLT bins below MLT 10h, we define an affine function linking the value of $L^*_{mp}$ at MLT 10h to the $L^*_{mp}$ at MLT 11h. This function is then extrapolated to other MLT bins, establishing the MLT dependence of the model as follows:

$$L^*_{mp}(MLT) = (m * MLT + b) * L^*_{mp}(MLT = 12) \tag{4}$$

With

$$m = 1 - \bar{L}^*_{mp} \quad (MLT = 10) \tag{5}$$

$$b = 1 - m * 11 \tag{6}$$

Figure 7 summarizes the model construction. The model is based on the $L^*_{mp}$ value at MLT 12h (in green). The flat distribution

described by the Eq. (1) (in yellow) extends the model to MLT 11h and 13h. The $Kp$ law, which defines the model's $L^*_{mp}$

value at MLT 10h (in red), is specified by the Eq. (2) and (3). The affine function (in blue) defining the $L^*_{mp}$ value across MLT

bins ranging from 6h to 10h is given by Eq. (4) to (6). As a result, the $L^*_{mp}$ at all dayside MLT bins can be expressed as a

function of the $L^*_{mp}$ at MLT 12h and the $Kp$ value, as illustrated by Figure 6.

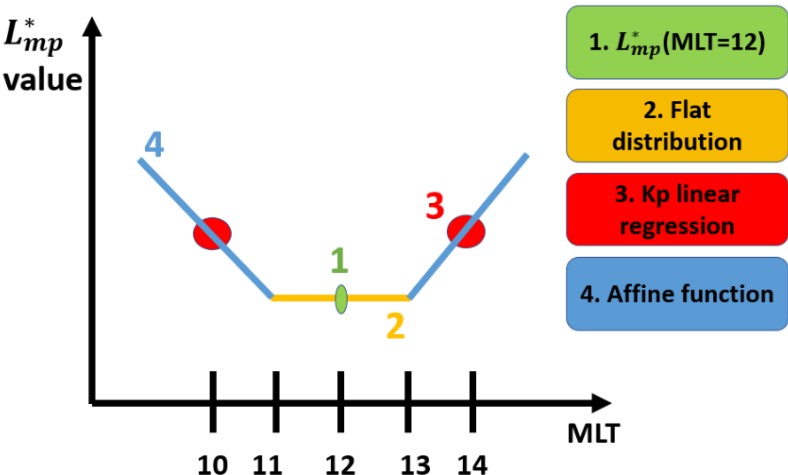

Figure 7 : Diagram describing the model construction process, with the $L^*_{mp}$ value at MLT 12h (green) propagated to MLT 11h and
13h (yellow), the $Kp$ linear regression giving the $L^*_{mp}$ value at MLT 10h and 14h (red) and the affine function extending the model
to other MLT bins (blue).

### 2.4 Assessing Model Performances

To assess the performances of our model, we compared its accuracy against the approach outlined in Herrera et al. (2016).

This approach is referred here to as the "3D Model." By contrast, the "4D Model" explicitly accounts for our new MLT-

dependent magnetopause location model. The accuracy of our model was evaluated by comparing its $L^*_{mp}$ calculations at each

dayside MLT sectors with the values in the 20-year dataset on which it is based. The performance was quantified using the

Root Mean Square Error (RMSE), which provides an interpretable measure of the average deviation between the model's

predictions and the dataset values.

Figure 8 presents the RMSE distribution of the 4D model and the 3D model for MLT 6h to 18h. The RMSE value of the 4D

model, between MLT 10h and 14h, remains around 0.1, reaching a minimum of zero at MLT 12h. This result aligns with the

model's design, which assumes a plateau extending the $L^*_{mp}$ value of MLT 12h two MLT hours around noon. Beyond the MLT 10h-14h range, the RMSE increases, reflecting the influence of the linear regression and affine function applied to MLT bins below MLT 10h.

Furthermore, at L* shells primarily influenced by magnetopause shadowing (L* between 6 and 8), the ΔL* ranges from 0.5 to 0.63 (Figure 1). In Figure 8, when the RMSE value is below 0.5, the error remains below the maximal intrinsic uncertainty of the Salammbô L* grid. Therefore, such errors do not impact significantly the modeling. For the 4D model, the RMSE stays under 0.5 for MLT values between 8h to 16h, demonstrating good accuracy. In comparison, the 3D model maintains an RMSE below 0.5 only for MLT bins between 10h and 14h, thus inducing complementary uncertainties.

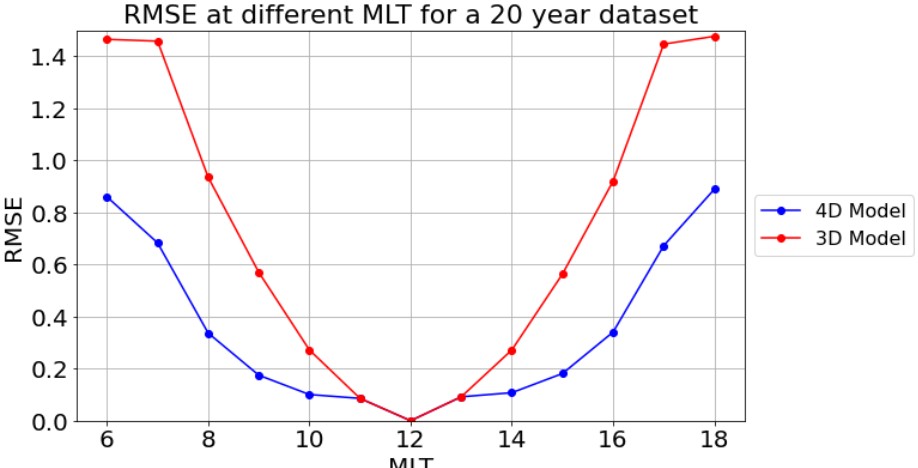

Figure 8 : Comparison of the RMSE of the new magnetopause location model (called 4D Model, in blue) and the RMSE of the $L^*_{mp}$ value at MLT 12h propagated at every MLT (called the 3D Model, in red) as a function of MLT.

Moreover, the RMSE values of the 3D model increases as the MLT moves away from MLT 12h, indicating that its fixed $L^*_{mp}$ value at MLT 12h does not generalize well across other MLT bins. In contrast, the 4D model provides a more accurate representation of the dataset across all MLT bins and geomagnetic activity levels. This improvement stems from its direct dependency to the MLT and the $Kp$ index and indirect dependence on solar wind parameters through $L^*_{mp}$ at MLT 12h.

**2.5 Qualitative evaluation of the model on the 2015 Saint Patrick Storm case study**

We then compare our semi-analytical model with the long-term dataset using the March 2015 Saint Patrick's storm (Baker et al., 2016; Goldstein et al., 2017; Li et al., 2016) as a case study. This analysis seeks to provide a qualitative assessment of the model's behavior during a geomagnetic storm.

The results of the comparison between the model estimations and the effective values are presented in Figure 9. We provide plots of $L^*_{mp}$ for MLT 13h (panel (a)) and MLT 15h (panel (b)) as function of time. In these plots, the blue dots represent our dataset, while the new model is depicted by the orange dotted line and the 3D model is displayed as the solid green line. The

plot in panel (a), at MLT 13h, confirms the flat distribution of $L_{mp}^*$ values between MLT 11h and 13h in the model's construction. Panel (b) demonstrates the 3D model's difficulty to reproduce the dataset values at MLT 15h.

While the 4D model generally performs well, there is a slight loss of accuracy in reproducing higher $L^*$ values on 15/03/2015 at 10 AM and 16/03/2015 at 10 PM. During these times, the difference between the data and the model is shown to be less than 0.25. The difference between the dataset's $L_{mp}^*$ value, calculated using $D_p$ and $B_z$, and the model's $L_{mp}^*$, based on the $Kp$ law, can be attributed to fluctuations during periods of low geomagnetic activity. During these times, the values of $D_p$ , $B_z$, and $Kp$ can vary significantly. This variability makes it challenging to accurately model the magnetopause using only the $Kp$

parameter. At $L^*$ values ranging from 7 to 8, Figure 1 demonstrates that differences below 0.5 are not detected by the Salammbô 4D code. Consequently, these minor inaccuracies do not impact the overall modeling results.

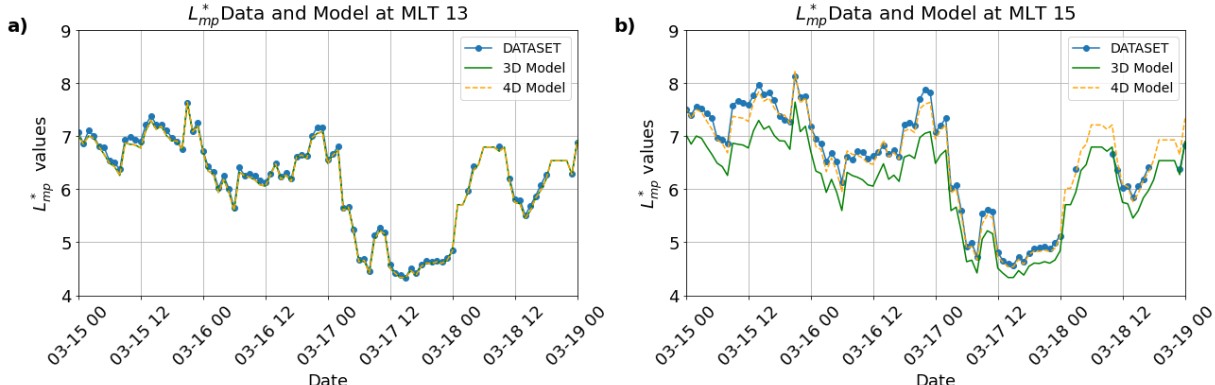

**Figure 9 : Comparison of the modelled $L_{mp}^*$ values of the new magnetopause location model (in orange dotted line), the modelled $L_{mp}^*$ value at MLT 12h (in solid green line) and the $L_{mp}^*$ value of the dataset used to create the model at MLT 13h (panel (a)) and**
**15h (panel (b)) during the March 2015 Saint Patrick storm.**

The new model demonstrates strong performance during the peak of geomagnetic storms, when the magnetopause is closest to Earth. This is a critical condition for accurate radiation belt modeling. A notable advantage of this semi-analytical model is its ability to reliably generate $L_{mp}^*$ values even in data-scarce environments. For example, during the data gap on March 18, 2015, between 12 AM and 12 PM, the model produced values consistent with neighboring observations. This highlights its
capability to provide coherent and reliable estimates in the absence of direct measurements.

## 3. Model Evaluation Against Magnetopause Crossing Data

The Shue model was originally developed using a magnetopause crossing database compiled up to 1998. Since then, several missions have been launched, providing additional magnetopause crossing data. These recent satellite crossings may be used to validate the new magnetopause model. For this purpose, we choose the catalog of magnetopause crossings of Michotte de
Welle et al.(2022) and Nguyen et al. (2022).This catalog uses satellite magnetopause crossings from the Double Star, MMS, Cluster and ARTHEMIS (THEMIS B and C) missions. The catalog provides extensive MLT coverage, including the dayside,

dawn, and dusk sides (as well as high latitudes with Cluster). For each spacecraft, the available data includes: epochs and the GSM coordinates position of the spacecraft, the ion bulk velocity components, $V_x$, $V_y$, $V_z$, the magnetic fields and its components, $B_x$, $B_y$, $B_z$, the ion density $N_p$ and the temperature $T$.

To enable comparison between the new $L^*$ model and the magnetopause crossing catalog, adjustments were made. As the magnetopause position model is equatorial, crossings were filtered out to include only those occurring at $|Z| \leq 1$ in GSM coordinates, focusing on the magnetic equatorial plane. Additionally, the satellite's crossing locations were transformed into $L^*$ values, aligning them with the format used in our model for direct comparison.

Figure 10 displays the scatter plot of crossing locations, with the color of the points reflecting the percentage of difference

between $L^*_{mp}$ values derived from the catalog and those predicted by the 3D model (panel(a)) and the 4D model (panel(b)). Panel (a) highlights that the differences between the 3D model and satellite crossings are smaller by up to 15 % around MLT 12 compared to other MLT bins, as expected. This observation supports the accuracy of the magnetopause location model proposed by Herrera et al. (2016) for MLT 12h. Panel (b) demonstrates that the 4D model achieves better accuracy across all other MLT bins compared to the 3D model, validating the improvement introduced by the MLT and $Kp$ dependencies in our

model.

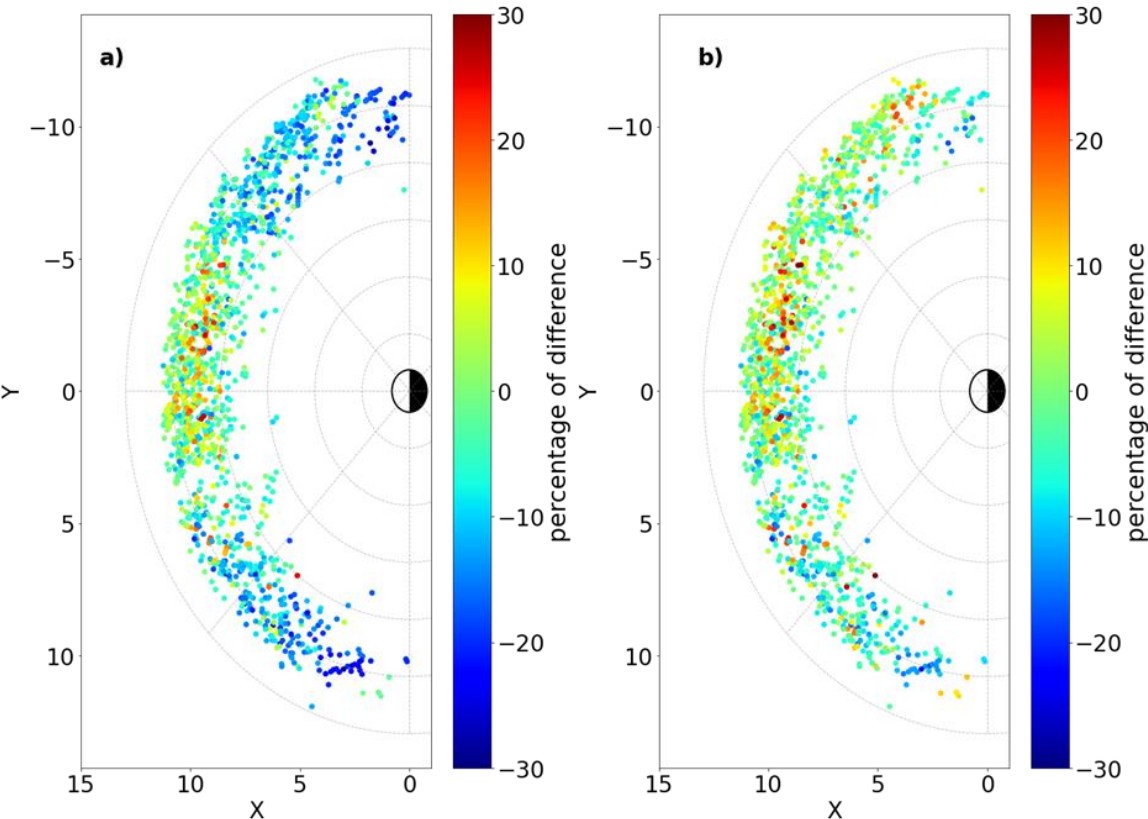

**Figure 10 : Side-by-side comparison of the relative differences between the satellite's $L^*_{mp}$ value and the $L^*_{mp}$ value from the 3D model (panel (a)), and the relative differences between the satellite's $L^*_{mp}$ value and the $L^*_{mp}$ value from the 4D model (panel (b)). The grid represents dayside MLT bins, with the Y=0 axis aligned along the Sun-Earth line. The color of each scatter point corresponds to the percentage of difference between the modelled $L^*_{mp}$ value and the satellite crossing $L^*_{mp}$ value, as indicated by the color bar.**

To evaluate the performance of the 4D model compared to the 3D model, we employed RMSE as the evaluation metric. Figure 11 shows a bar plot comparing the RMSE values between the magnetopause crossing catalog and the two models. The 4D model (in red) exhibits a consistently lower RMSE than the 3D model (in blue) across all MLT values except MLT 10h. At MLT 10h, the modeling of $L^*_{mp}$ relies on a $Kp$-dependent linear regression and on an affine function extending the model at other MLT bins. Notably, it demonstrates a lower RMSE in the dawn sector, aligning with the improved accuracy in this region as also suggested by Figure 10. Overall, the 4D model's lower RMSE indicates superior performance in accurately describing the MLT-dependent variation of the magnetopause location. The absence of an RMSE value at MLT 18h is explained by the low number of points used for its calculations in this sector.

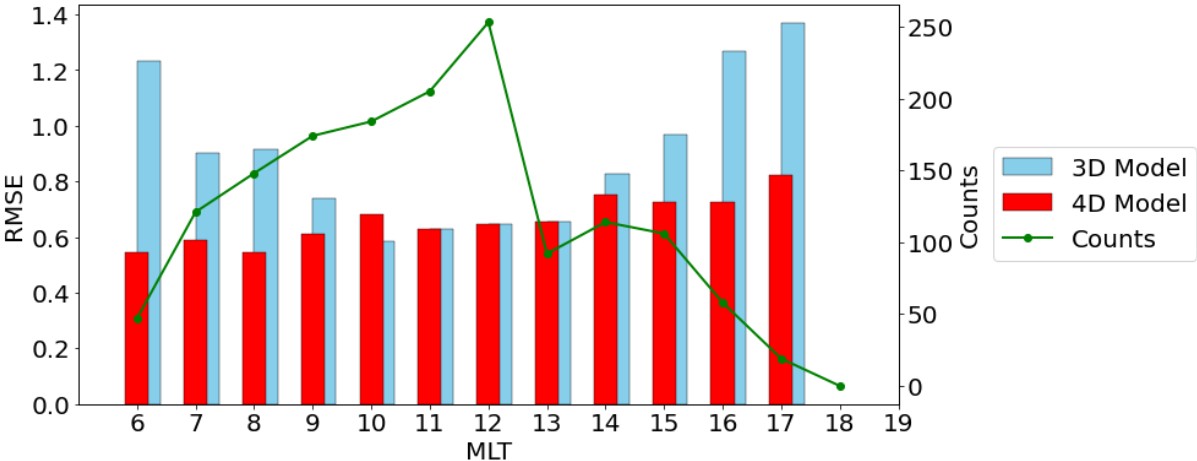

Figure 11 :Comparison of RMSE values (left y-axis) for the new 4D model (red) and the 3D model (blue) as a function of MLT, along with the number of points used to calculate the RMSE (right y-axis).

The symmetrical nature of the Shue magnetopause model, used as the reference for constructing the dataset, resulted in a symmetrical 4D magnetopause location model. Figure 11 demonstrates that the model has a better accuracy on the dawn sector than in the dusk sector. The dawn-dusk asymmetry of the magnetopause during disturbed times documented by Dmitriev et al. (2004) suggests that the reliance on a symmetric reference model may have introduced inherent limitations.

Validating the 4D model against the recent magnetopause crossing catalog relies on the assumption that the Shue model remains accurate for all conditions. At MLT 12h, this validation indirectly evaluates the Shue model itself. Specifically, a comparison between the predicted distances from the Shue model (converted in $L^*$) and recent satellite crossings (also in $L^*$), resulted in an RMSE value of 0.65, indicating the existence of inherent uncertainties in the Shue model. Staples et al. (2020) investigated the Shue model's accuracy using satellite data and found it to be reasonably reliable during periods of low geomagnetic activity. However, its reliability decreased under disturbed conditions, with an error margin of $\pm$ 1 $R_E$. While their study focused on distances in $R_E$, it supports the utility of the Shue model but also highlights potential uncertainties arising from its limitations. These limitations should be considered when interpreting the 4D model's performance, ensuring that any residual discrepancies are understood in the context of the underlying assumptions rather than as flaws in the 4D model itself.

The validation confirms that our new magnetopause location model provides an accurate representation of the magnetopause $L^*$ location. Since this model serves as a boundary for the $L^*$ parameter in the Salammbô code, its influence is most pronounced on the outermost $L^*$ shells of the Salammbô grid. At high $L^*$ values, Figure 1 shows that the $L^*$ grid has low refinement, with a spacing of 0.63 between $L^* = 8$ and $L^* = 7.3$. The 4D model's RMSE values, ranging from 0.54 at MLT 6h to 0.82 at MLT 17h, generally fall within this grid spacing. Therefore, the model's error is unlikely to significantly affect the radiation belt modeling in the Salammbô code. Even when the RMSE slightly exceeds 0.63, the resulting impact would be minimal, leading to a shift of at most one grid cell in a Salammbô simulation scheme.

## 4. Impact of the new magnetopause model in the Salammbô 4D code: The "St. Patrick's Day" storm in March 2015

The Salammbô 4D code is a modified version of the Salammbô code of Bourdarie et al. (1996) that accounts for the dynamics

of low energy electrons. The boundary condition is derived from THEMIS data (Maget et al., 2015) to which a gaussian distribution as a function of MLT centered around midnight has been added to account for this added dimension. The code considers the same radial diffusion model used in Boscher et al. (2018) and Brunet et al. (2023). The magnetospheric electric field model used to model the convective transport of particles is the UNH-IMEF Matsui et al. (2013) numerical model. Wave particle interactions are taken into account using estimations pitch angle and energy diffusion coefficients from the WAPI code

Sicard-Piet et al. (2014) using an MLT dependent classification of VLF waves. Finally, rapid losses from magnetopause shadowing are implemented such that: all particles with an L* greater than L* defined by the magnetopause location model are lost.

A comparison between the new and the previous magnetopause models has been made through the simulation of the March 2015 Saint Patrick storm. Results, depicted in Figure 2, showcase the electron phase space density (PSD) on March 17, 2015,

at 4:45 am, few times before the storm's peak. Panel (a) depicts the behavior of the 3D model with a MLT-constant $L^*$ value for the magnetopause location equal to $L^*_{mp}$ at MLT 12h. Panel (b) illustrates the behavior of our novel magnetopause model that considers the MLT dependency and allows a more gradual magnetopause shadowing process on the dayside. The ratio between the new model and the previous one can be seen on panel (c), where the PSD ratio globally higher than 1 indicates that more particles are lost with the 3D model than with the 4D one. Although the magnetopause shadowing is only applied to

the dayside, ratios are also different from 1 in the nightside. This is due to the electric fields that induce a convective transport of particles (both in MLT and in L*). Panel (c) highlights the importance of an MLT-dependent magnetopause model, where particles lost in the 3D model are transported to the nightside in the 4D model due to this convective transport mechanism. Figure 3 supports this assumption by showing the omni-directional fluxes (FEDO) at the magnetic equator for 100 keV electrons as a function of MLT on March 17, 2015, at 4:45 AM, based on two Salammbô 4D simulations. The first one uses

the 3D magnetopause location model (in blue) while the other one considers the 4D magnetopause location model (in red). Panel (a) illustrates the fluxes of electron at $L^* = 5.4$ where particles get impacted by the magnetopause shadowing and panel (b) shows the fluxes of electron at $L^* = 5$ on a closed drift shell. Both panels illustrate that the 3D model has lower fluxes compared to the 4D one. This means more particles are lost with the 3D model. In contrast, our 4D model preserves these particles, which are transported deeper in the radiation belts. Consequently, we can state that the use of the 3D magnetopause

model on a 4D radiation belt code results in excessive loss of low energy particles compared to our new magnetopause model that is MLT dependent.

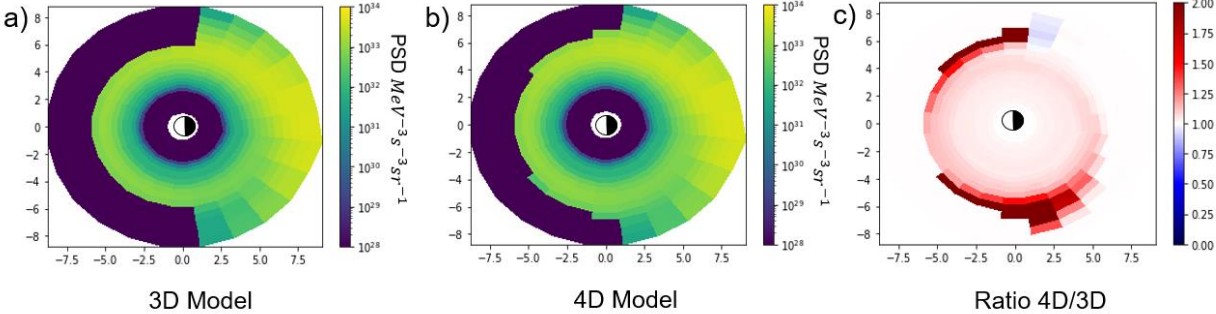

|3D Model|4D Model|Ratio 4D/3D|

**Figure 12: comparison between the phase space density of a Salammbô simulations at $\mu = 5\ MeV/G$ during the geomagnetic storm of march 2015 with the 3D magnetopause model (panel (a)), the 4D magnetopause mode (panel (b)) and the ratio of the two (panel (c))**

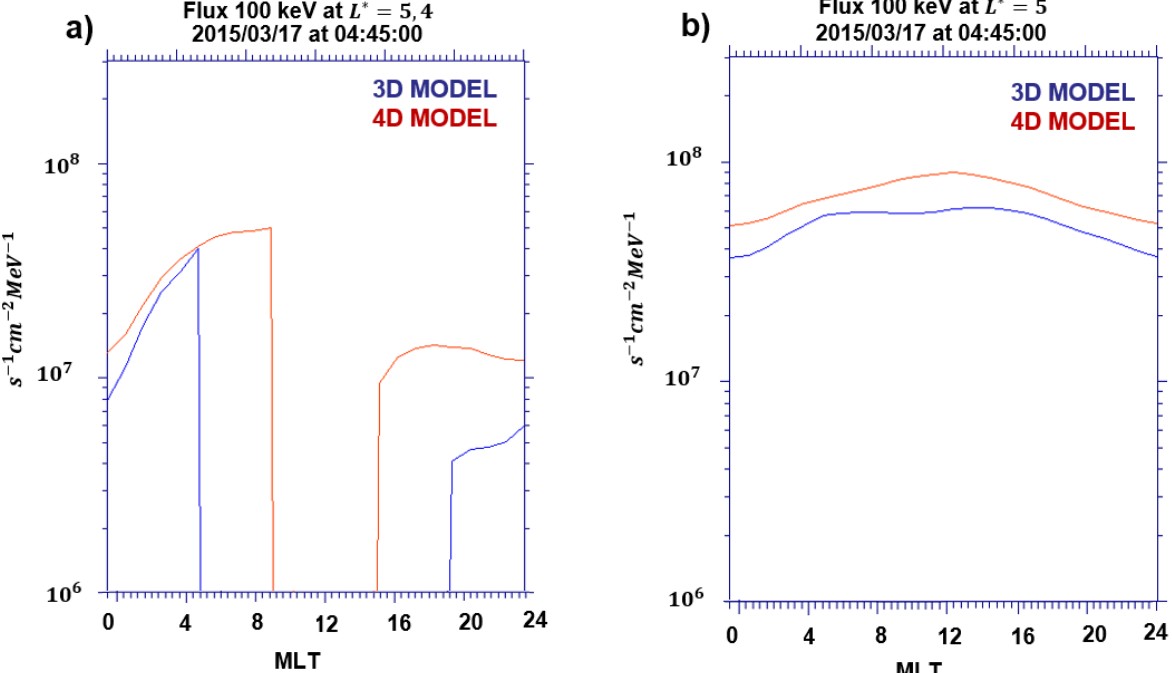

**Figure 13: Comparison of FEDO fluxes at the magnetic equator for 100 keV electrons as a function of MLT at a fixed time step during the March 2015 geomagnetic storm, simulated using the Salammbô model. The fluxes are shown for the 3D magnetopause location model (blue) and the new 4D magnetopause location model (red). Panel (a) shows the fluxes at $L^* = 5.4$ and panel (b) at $L^* = 5$.**

**Conclusion**

This study presents a novel semi-analytical model expressing magnetopause location as a function of $L^*$ and MLT. This model is specifically designed for MLT-dependent convective and diffusive radiation belt modeling codes, which require the
magnetopause position expressed in terms of magnetic parameters to account for the magnetopause shadowing effect. Unlike previous models such as those by Shue et al. (1998) and Lin et al. (2010), which define the magnetopause in Earth radii (Re), our model provides a more suitable representation for the inner magnetosphere by using $L^*$ as a key coordinate. Building on Herrera et al. (2016) 3D model at MLT 12h, our approach extends coverage to other dayside MLT sectors.

Our model is based on a statistical analysis of $L^*$ calculations and was validated against a detailed satellite magnetopause
crossing catalog. The results demonstrate accuracy in estimating the magnetopause position in $L^*$, particularly in the dawn sector, thanks to our model's dependence on MLT and the $Kp$ index. Our approach shows that an appropriate modeling is one that represents the magnetopause as rather blunt on the dayside but with a clear $Kp$ dependence in other MLT sectors.

The integration of our magnetopause location model into Salammbô 4D radiation belt model will improve its accuracy. This improvement relies on the $L^*$ grid resolution being sufficiently refined to capture particle density variations due to
magnetopause shadowing, while also minimizing model errors. Validation against a comprehensive magnetopause crossing catalog confirms the model's reliability. Moreover, the model addresses the challenge of lack of data, providing a continuous estimation of the magnetopause location. This continuity makes it an excellent fit for radiation belt modeling, where gap-free boundary conditions driving input data are essential. The first integration of the new magnetopause model into the Salammbô 4D code highlights its coupling with the physical processes in the radiation belt dynamic modeling, particularly the convective
transport of particles driven by magnetospheric electric fields.

This equatorial magnetopause location model is deliberately tailored for incorporation into MLT-dependent radiation belt codes at the magnetic equator, following (Herrera et al., 2016) justification (section 2). We prioritized in this work $L^*$ and MLT dependency to extend (Herrera et al., 2016) model. We acknowledge that a more advanced, non-equatorial magnetopause location model could be explored in future research.

Future studies should focus on validating the model through statistical comparisons with in-situ satellite data and the satellite trajectories mapping in terms of $L^*$ and MLT, incorporated into Salammbô simulations. This approach will enhance our understanding of particle loss processes associated with magnetopause shadowing in the radiation belts.

**Author contribution**

R. K. developed the model under the guidance and supervision of N. D. and V. M. Statistical analysis and interpretation of results have been collaboratively supported by V. M., B. L. and N. D.. R. K. generated the figures and drafted the paper with the contribution of all co-authors.

## Competing interests

The authors declare that they have no conflict of interest.

## Acknowledgements

We are thankful to NASA's OMNI for online data access. We are also thankful to G. Nguyen, B. Michotte De Welle and A. Ghisalberti for granting us access to their work on the satellite magnetopause crossing catalog.

## Financial Support

This study has been funded by ONERA – The French Aerospace Lab.

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
