# Peer review of "Modeling magnetopause location for 4D drift-resolved radiation belt codes: Salammbô model implementation."

_EGUsphere, 2024_

## Author Response (AR1)

**Author's Response to Reviews of**

*Modeling magnetopause location for 4D drift-resolved radiation belt codes: Salammbô model implementation.*

Rabia Kiraz, Nour Dahmen, Vincent Maget, Benoit Lavraud

**Response to referee comments from Reviewer 1**

We would like to thank the referee for the constructive feedback on this article. We have strived to address all the remarks and comments, for which we report our detailed response below.

*This study built an MLT-dependent magnetopause location model expressed in terms of $L^*$ and driven by the Kp index. This model can be used in the ring current and radiation belt simulations. However, this model doesn't consider drift shell splitting and has no dependence for the second adiabatic invariant, which has been demonstrated to be important in simulations of radiation belt dropouts. Several concerns need to be addressed before I can recommend this manuscript for publication.*

1. *Due to the drift shell splitting, $L^*$ values for particles measured at the same locations with different pitch angles are distinct. However, the proposed magnetopause model does not include pitch angle or second invariant (K) dependence. Previous studies (Tu et al., 2019) have demonstrated that K-dependent last closed drift shell is critical to simulating radiation belt dropout. The authors should discuss how the absence of K-dependent in the model might affect its accuracy.*

Our magnetopause model is an equatorial model that indeed does not consider the second invariant. In fact, we followed the method developed in Herrera et al., (2016), which justifies the utility of an equatorial model. Their justification acknowledges the implications of neglecting non-equatorial dynamics, as noted by the reviewer. Particles bouncing outside the magnetic equator along a field line at noon tend to drift along higher L shells, increasing their likelihood of being untrapped compared to equatorial mirroring particles.

We propose to add the following paragraph to the paper in line 353:

This equatorial magnetopause location model is deliberately tailored for incorporation into MLT-dependent radiation belt codes at the magnetic equator, following Herrera et al. 2016 justification (section 2). We prioritized in this work $L^*$ and MLT dependency to extend Herrera et al., (2016) model. We acknowledge that a more advanced, non-equatorial magnetopause location model could be explored in future research.

We also propose to add precisions in the database construction section (line 137) that the transformations to $L^*_{mp}$ from the Shue model are performed at the magnetic equator.

2. *Line 94-95. Xiang et al. (2017) is the first study to suggest that LCDS is more reliable than magnetopause standoff position to evaluate the impact of magnetopause shadowing.*

Line 94-95 will be amended with the addition of the reference of Xiang et al. (2017).

3. *Line 115. The effects of magnetospheric electric fields on particles are emphasized to support the necessity of building a 4D radiation belt modeling. However, the proposed magnetopause model seems not account for the effects of these magnetospheric electric fields.*

This magnetopause location model aims to locate the magnetopause in a geomagnetic reference frame using the notion of a drift shell. Therefore, it is a purely magnetic magnetopause model. Electric field modeling is also important to consider, particularly at low energies. The effect of the electric field is of course also modelled in the Salammbô code. The paper focuses on the benefits of this new magnetopause location model, making it unnecessary to study the impact or complexity of the electric field for a first implementation and validation.

4. *Figure 10, Figure 11. These two figures demonstrate that the 4D model has better performance than the 3D model. However, the differences between 4D model and satellite observations at some locations have large values (>2). The RMSE is always >0.5 at all MLTs. This raises questions about the model's accuracy in representing the magnetopause L\* location. The authors should provide a more detailed discussion of these discrepancies*

Indeed, we acknowledge the presence of a bias which is introduced by using the Shue model. Staples et al., (2020) demonstrated an error of $\pm 1$ $R_E$ when comparing the Shue model to a large database of magnetopause location in $R_E$. We replicated this comparison using the magnetopause crossing catalog of Nguyen et al., (2022) in the geographic reference frame. Figure 1 below depicts the statistical error of the Shue model as a function of MLT bins. It reveals a statistical error at MLT 12h of approximately 0.6 $R_E$. This error is propagated when transformed into a $L^*$, contributing to an RMSE >0.5 in our model (Figure 11, line 315). For context on the minimal impact of this error within the Salammbô 4D code, see our discussion in lines 331-335. We propose adding Staples' numerical precision of the Shue model error (originally mentioned in line 323) for completeness.

[Figure]

*Figure 1 : RMSE of the Shue model's radial distances compared to radial distances of the magnetopause crossing catalog.*

5. *The new magnetopause locations model is developed for the 4D Salammbo model. However, there is not any simulation results with the new model in the study. Some comparisons between the simulation results using the new and previous magnetopause location model in the 4D Salammbo model are needed to validate the advances of the new model.*

Reviewer 2 also had the same comment and we acknowledge that this part was missing in our initial paper. To that effect we incorporated an additional section to our paper, dedicated to compare the impact of both models on Salammbô simulations. To illustrate this comparison, we leveraged a case study of the Saint Patrick geomagnetic storm of March 2015, presenting side-by-side figures of two

Salammbô simulation runs: one employing the traditional 3D magnetopause model and the other utilizing our new model. See new section IV below.

IV. Impact of the new magnetopause model in the Salammbô 4D code: The "St. Patrick's Day" storm in March 2015

The Salammbô 4D code is a modified version of the Salammbô code of Bourdarie et al. (1996) that accounts for the dynamics of low energy electrons. The boundary condition is derived from THEMIS data (Maget et al., 2015) to which a gaussian distribution as a function of MLT centered around midnight has been added to account for this added dimension. The code considers the same radial diffusion model used in Boscher et al. (2018) and Brunet et al. (2023). The magnetospheric electric field model used to model the convective transport of particles is the UNH-IMEF Matsui et al. (2013) numerical model. Wave particle interactions are taken into account using estimations pitch angle and energy diffusion coefficients from the WAPI code Sicard-Piet et al. (2014) using an MLT dependent classification of VLF waves. Finally, rapid losses from magnetopause shadowing are implemented such that: all particles with an L* greater than L* defined by the magnetopause location model are lost.

A comparison between the new and the previous magnetopause models has been made through the simulation of the March 2015 Saint Patrick storm. Results, depicted in Figure 2, showcase the electron phase space density (PSD) on March 17, 2015, at 4:45 am, few times before the storm's peak. Panel (a) depicts the behavior of the 3D model with a MLT-constant $L^*$ value for the magnetopause location equal to $L_{mp}^*$ at MLT 12h. Panel (b) illustrates the behavior of our novel magnetopause model that considers the MLT dependency and allows a more gradual magnetopause shadowing process on the dayside. The ratio between the new model and the previous one can be seen on panel (c), where the PSD ratio globally higher than 1 indicates that more particles are lost with the 3D model than with the 4D one. Although the magnetopause shadowing is only applied to the dayside, ratios are also different from 1 in the nightside. This is due to the electric fields that induce a convective transport of particles (both in MLT and in L*). Panel (c) highlights the importance of an MLT-dependent magnetopause model, where particles lost in the 3D model are transported to the nightside in the 4D model due to this convective transport mechanism.

Figure 3 supports this assumption by showing the omni-directional fluxes (FEDO) at the magnetic equator for 100 keV electrons as a function of MLT on March 17, 2015, at 4:45 AM, based on two Salammbô 4D simulations. The first one uses the 3D magnetopause location model (in blue) while the other one considers the 4D magnetopause location model (in red). Panel (a) illustrates the fluxes of electron at $L^* = 5.4$ where particles get impacted by the magnetopause shadowing and panel (b) shows the fluxes of electron at $L^* = 5$ on a closed drift shell. Both panels illustrate that the 3D model has lower fluxes compared to the 4D one. This means more particles are lost with the 3D model. In contrast, our 4D model preserves these particles, which are transported deeper in the radiation belts. Consequently, we can state that the use of the 3D magnetopause model on a 4D radiation belt code results in excessive loss of low energy particles compared to our new magnetopause model that is MLT dependent.

[Figure]

| 3D Model | 4D Model | Ratio 4D/3D |

*Figure 2: comparison between the phase space density of a Salammbô simulations at $\mu = 5\ MeV/G$ during the geomagnetic storm of march 2015 with the 3D magnetopause model (panel (a)), the 4D magnetopause mode (panel (b)) and the ratio of the two (panel (c))*

[Figure]

*Figure 3: Comparison of FEDO fluxes at the magnetic equator for 100 keV electrons as a function of MLT at a fixed time step during the March 2015 geomagnetic storm, simulated using the Salammbô model. The fluxes are shown for the 3D magnetopause location model (blue) and the new 4D magnetopause location model (red). Panel (a) shows the fluxes at $L^* = 5.4$ and panel (b) at $L^* = 5$.*

**Response to referee comments from Reviewer 2**

We would like to thank the referee for the constructive feedback on this article. We have strived to address all the remarks and comments, for which we report our detailed response below.

*This paper developed a magnetopause location model which considered the drifting effects based on statistical analysis of observational data. The produced magnetopause model is dependent on L\*, MLT, and Kp index. The model is an important component to global Fokker-Planck models, providing constrains on the simulation boundaries and magnetopause shadowing. The semi-analytical format of the model also makes it an efficient model. This paper studied an important subject and is recommended to be published after the following minor issues are resolved.*

1. *In Figure 5, the author used a linear fit for the Kp dependence at MLT of 10. The curve shown, however, seems closer to a parabolic fit. Can the author explain why a linear fit is used instead of any higher order fit?*

In our opinion, a parabolic fit that curves upward is unrealistic, as bump appears artificial and the plateau at high Kp is likely due to both physical factors and statistical limitations. This is why a linear fit appeared more physically meaningful for us. Nevertheless, during the model construction we tested a higher-order fit. Figure 4 illustrates the parabolic fit (dashed green line) and the linear regression fit (dashed red line) used for describing the evolution of $\overline{L^*}_{mp}$ at MLT 10H as a function of Kp. The comparison between the two fits in figure 5, shows that using a higher order fit has no significant impact on the final RMSE of the model. Therefore, for the sake of simplicity, we opted for a linear fit. We do not believe that further discussion on this topic is necessary to be added in the paper.

[Figure]

*Figure 4: L values of the magnetopause for different MLT values as a function of Kp with a linear regression fit (dashed red line) and a parabolic fit (dashed green line).*

[Figure]

*Figure 5: RMSE comparison between a polynomial fit and a linear fit in the model construction.*

2. *Line 93: "modelmodeling" is a typo.*

Line 93 will be amended with the suppression of the typo.

*3. Line 318 "The documented dawn-dusk asymmetry of the magnetopause during disturbed times.": This sentence is incomplete.*

Line 318 will be amended with the completion of the sentence.

*4. Figure 10: The color-bar makes it hard to distinguish the performances of 3D and 4D models. It is suggested that the author changes a color-bar or chooses a different format for the figure.*

Figure 10 will be amended with a new color-bar colormap and a new color-bar unit with relative ratio in percentages.

*5. The author provided many discussions on the importance of a MLT dependent magnetopause model in the FP simulations. However, it is not shown or discussed, how the new 4D model will affect the simulations, aside from reducing the boundary error to below the grid sizes. The author should present some discussions, or preferably simulation results with the new 4D model.*

Reviewer 1 had the same comment and we acknowledge that this part was missing in our initial paper. To that effect we incorporated an additional section to our paper. This new section focuses on comparing the utilization of 3D versus 4D magnetopause models in Salammbô simulations. To illustrate this comparison, we leveraged a case study of the Saint Patrick geomagnetic storm of March 2015, presenting side-by-side figures of two Salammbô simulation runs: one employing the traditional 3D magnetopause model and the other utilizing our new model. See new section IV below.

[revised manuscript text omitted]

---

## Author Response (AR2)

**Author's Response to Reviews of**

*Modeling magnetopause location for 4D drift-resolved radiation belt codes: Salammbô model implementation.*

Rabia Kiraz, Nour Dahmen, Vincent Maget, Benoit Lavraud

**Response to referee comments from Reviewer 1**

*The authors addressed most of my concerns. I only have a minor suggestion for the Figure 2 and Figure 3 in the response letters. The comparison of simulation results between the new and the previous magnetopause models during the March 2015 storm are shown in Figure 2 and Figure 3. Can the authors also add satellite observations in the two figures to support that the new magnetopause model is a better choice in the simulation.*

We sincerely thank the reviewer for this remark. Comparing Salammbô simulations with satellite measurements is a challenging task. For the study of the Saint Patrick Storm event in March 2015, we consulted our database to identify available satellite data. The relevant satellites are RBSP A and B, THEMIS A, D, and E, GOES 13 and 15, as well as LANL 01, 02, 04, 97, 91, and 94. However, only GOES 13, LANL 97, and LANL 91 provided usable data. Consequently, our analysis is limited to geosynchronous orbit, restricting the validation of our magnetopause location model. Due to the scarcity of suitable satellite data, a comprehensive validation against in situ measurements remains an open task. While this aspect requires further investigation, we believe that, at this preliminary stage, validation based on a magnetopause crossing catalog provides a reasonable first step. Future studies should focus on a more extensive validation of the magnetopause location model using satellite data from multiple geomagnetic storms.